# PUL-Mediated Plant Cell Wall Polysaccharide Utilization in the Gut Bacteroidetes

**DOI:** 10.3390/ijms22063077

**Published:** 2021-03-17

**Authors:** Zhenzhen Hao, Xiaolu Wang, Haomeng Yang, Tao Tu, Jie Zhang, Huiying Luo, Huoqing Huang, Xiaoyun Su

**Affiliations:** State Key Laboratory of Animal Nutrition, Institute of Animal Sciences, Chinese Academy of Agricultural Sciences, Beijing 100193, China; zhenzhenhao2012@163.com (Z.H.); xiaolu4444@126.com (X.W.); yhmbjbj@126.com (H.Y.); tutao@caas.cn (T.T.); zhangjie09@caas.cn (J.Z.); luohuiying@caas.cn (H.L.)

**Keywords:** plant cell wall polysaccharides, polysaccharide utilization systems, Bacteroidetes, gut microbiota

## Abstract

Plant cell wall polysaccharides (PCWP) are abundantly present in the food of humans and feed of livestock. Mammalians by themselves cannot degrade PCWP but rather depend on microbes resident in the gut intestine for deconstruction. The dominant Bacteroidetes in the gut microbial community are such bacteria with PCWP-degrading ability. The polysaccharide utilization systems (PUL) responsible for PCWP degradation and utilization are a prominent feature of Bacteroidetes. In recent years, there have been tremendous efforts in elucidating how PULs assist Bacteroidetes to assimilate carbon and acquire energy from PCWP. Here, we will review the PUL-mediated plant cell wall polysaccharides utilization in the gut Bacteroidetes focusing on cellulose, xylan, mannan, and pectin utilization and discuss how the mechanisms can be exploited to modulate the gut microbiota.

## 1. Introduction

Plant cell wall polysaccharides (PCWP), mainly consisting of cellulose, hemicellulose, and pectin, are among the most abundant renewable carbohydrates in nature [1,2]. The component distribution of PCWP varies by plant types; however, on average cellulose accounts for ca. 40% and hemicellulose (most commonly xylan) constitutes ca. 30% [3]. Cellulose is made up of tightly packed linear polymeric chains with 100 to 1000 β-(1,4)-linked D-glucose units [1]. In contrast, hemicellulose presents branched heterogenous structures composed of xylose-configured main chains decorated with side chains of much differing linkages (arabinose, acetate, ferulic acid, etc.) [3].

Efficient degradation and utilization of PCWP is of great value for biofuel [4,5,6,7], food [8,9,10], paper [11,12], and textile industries [11] and particularly important for the livestock industry since the depolymerized constituent sugars serve as the carbon and nutrient source for many gut microbes. These microbes in turn convert PCWPs to volatile fatty acids that influence host growth, physiology, and health. PCWP degradation requires orchestrated action of a multitude of carbohydrate-active enzymes (CAZymes), particularly of those classified as glycoside hydrolase (GH), carbohydrate esterase (CE), polysaccharide lyase (PL), and recently discovered polysaccharide monooxygenase (PMO) [13,14]. Synergy of these enzymes is exemplified by three strategies which are (i) free enzymes, typified by the aerobic filamentous fungus *Trichoderma reesei* [15]; (ii) cellulosome, best known in the anaerobic bacterium *Clostridium thermocellum* [16]; and (iii) an intermediate form represented by a multi-modular bifunctional enzyme CelA found in the *Caldicellulosiruptor bescii* [17,18,19]. As an addition to these, it is important to note that maximal PCWP degradation may involve both enzymes and microbes, in which the microbe is attached to specific polysaccharides [20,21].

Attaching to the polysaccharides creates proximity for the microbe to the substrate, which enables cooperation between the enzymes and the microbes. This is because the destructed sugars are rapidly assimilated by the cell, alleviating the product-induced feedback inhibition of enzymes. Microbes with such a binding feature include members of Bacteroidetes, which encode CAZymes with type II signal peptides, allowing the enzymes to attach to the outside of the cell membrane. Interestingly, many of these enzymes have carbohydrate-binding modules (CBMs) or other polysaccharide-binding domains, enabling them to bind to specific polysaccharides [22]. The Bacteroidetes constitute a dominant phylum in the intestinal tract of humans and livestock animals and these bacteria metabolize a broad scope of dietary and host-derived polysaccharides, including the relatively simple glycans (such as starch [23], levan [24], and inulin) as well as cellulose and complex hemicellulose polysaccharides (like xylan [25,26,27], arabinan [28], galactan [29], arabinogalactan [30], xyloglucan [31,32], and galactomannans [33]). Their genomes typically contain abundant GHs and PLs corresponding to the varying types of polysaccharides [22,34]. The recently reported *Bacteroides cellulosilyticus* WH2 encodes 503 CAZymes comprising 373 GHs, 23 PLs, 28 CEs, and 84 glycosyltransferases [35]. Of special note is the fact that the CAZyme-encoding genes are typically co-localized within the genome at certain polysaccharide utilization loci (PULs), which was first presented by Bjursell et al. [36] to describe clusters of co-localized and likely co-regulated genes. The products of these genes orchestrate in regulating, sensing, recognizing, binding, enzymatic digestion, and transporting of (degraded) complex polysaccharide substrates into periplasmic space and cytosol [22,37,38,39] (Figure 1).

The gut microbiota has tremendous effects on the health of humans and growth performance of livestock. Enzymes, particularly glycoside hydrolases, have a long history of being added to animal feed to improve the feed conversion rate [40,41,42]. However, it has not drawn much attention until recently that added enzymes change the gut microbiota. This provides a possibility of intended alteration of the gut microecology using exogenously added enzymes. Knowing how the gut-resident bacteria utilize plant cell wall polysaccharides is the basis towards designing enzymes useful in precisely modulating the gut microbiota. Here we will review the PUL systems of the dominant Bacteroidetes in the gut, focusing on PUL-mediated utilization of major components of the plant-derived polysaccharides by the bacteria.

## 2. The Bacteroidetes in the Gut

Bacteroidetes is one of the most abundant phyla in the gut microbial community of human and animal guts. This phylum includes mainly members of the Bacteroides, Prevotella, Alistipes, and Parabacteroides, with the former two being dominant. The Bacteroides species are Gram-negative, spore-free, obligate anaerobic bacillus. *Bacteroides fragilis*, a human pathogen associated with appendicitis, was the first Bacteroides species to be isolated [43]. It had been quite challenging to isolate intestinal anaerobic microbes including the Bacteroides species. However, with the improvement of anaerobic culture technology [44], more and more Bacteroides species were gradually isolated from feces and the gastrointestinal tract. Now it becomes clear that Bacteroidetes is a large phylum, with diversity in the genome at every level from genus to the strain, enabling the members of Bacteroides highly adaptable to rapidly changing intestinal environments [45]. Most of the members of Bacteroides live in the distal gut and can digest the polysaccharides derived from diet and even the host. By fermenting these polysaccharides into short-chain volatile fatty acids, they exert roles by providing nutrition to the host, maintaining the stability of the intestinal microecology, and even impacting profoundly the host’s immune system [46], obesity, and complex diseases [47,48]. Although a few members, such as *B. fragilis*, of the Bacteroidetes genus are harmful to the host under certain circumstances, gut microbes may be beneficial rather than harmful when a normal intestinal environment is maintained [49]. In this area, one major challenge is how to establish a metabolic or physiological link between the microbes and the host. The PUL systems of Bacteroidetes (with selected examples listed in Table 1) provide such a connection between the gut microbes and host.

## 3. PUL-Mediated Plant Cell Wall Polysaccharides Degradation

### 3.1. Cellulose Utilization System (Cellulose-PUL)

The term cellulose utilization system, associated with an uncultured Bacteroidales phylotype AC2a of cow rumen, has been put forward by Naas et al. [50,51]. Since then, cellulolytic PULs have been identified in Bacteroidetes originating from Svalbard reindeer rumen [52,53], cow and buffalo rumen [50,54], and tammar wallaby foregut [53]. The Bacteroidales Ac2a PUL contains eight genes which encode two putative GH5 and GH9 cellulases, a GH94 cellobiose phosphorylase [53], a SusC-like transporter, a SusD-like [55] and a SusE-positioned surface glycan binding protein (SGBP), an inner membrane sugar transporter, and an inner membrane sensor [28]. Cellulose is believed to be bound on the cell surface through the concerted action of SusD-like and SusE-positioned cell membrane SGBP proteins. GH5 and GH9 further cooperate to degrade cellulose into cellobiose, which is then transported via the SusC-like transporter into the periplasm. In periplasm, it is converted by the cellobiose phosphorylase into glucose and imported to the cytoplasm for cellular metabolism [50].

### 3.2. Xylan Utilization System (Xylan-PUL)

Like cellulose, xylan is rich in animal feed as well as in the human diet. It cannot be used by humans or animals, but rather requires gut microbes for degradation [56]. Xylan has a backbone of β-1,4 linked xylopyranoses with side chains consisting of arabinose, α-glucuronic acid,acetate, ferulic acid, etc. [57,58]. Complete degradation of xylan, therefore, requires the synergistic action of a range of enzymes, including endo-β-1,4-xylanase and β-xylosidase to cleave the backbone, the arabinofuranosidase, α-glucuronidase, acetyl esterase, and feruloyl esterase to remove the side chains [59,60].

Although not all gut bacteria can degrade xylan, many Bacteroidetes members are found to be capable of degrading xylan, including *Bacteroides eggerthi* [61], *B. fragilis* [62,63], *B. cellulosilyticus* [64,65], *B. intestinalis* [25,66], *B. ovatus* [28,67,68], *B. xylanisolvens* [27,69], and *Prevotella bryantii* [70,71]. In these bacteria, the genes responsible for xylan utilization are most often discovered in XUS gene clusters. The core gene sets of different XUSs are highly conserved in different xylan-utilizing Bacteroides.

The *B. ovatus* strain ATCC 8483 can utilize both simple, linearized xylan and complex cereal glucuronoarabinoxylan (GAX). In accordance, its genome contains two PULs (large and small) targeting substrates with different complexity [68]. Both the large (BACOVA_03417-50, PUL-XylL) and small xylan PUL (BACOVA_04385-94, PUL-XylS) are activated when the bacterium is grown on wheat arabinoxylan (WAX) [28]. WAX is bound to a SGBP in the PUL-XylS (BACOVA_04391) on the cell surface and degraded by concerted action of a GH10 xylanase (BACOVA_04390) [25], a GH3 β-xylosidase (BACOVA_03419) [32], and a GH43 α-arabinofuranosidase (BACOVA_03421). The arabino-xylooligosaccharides (AXOS) products are imported into the periplasm via the SusC/SusD-like complex, and further broken down by two GH43 α-arabinofuranosidases (BACOVA_03425 and BACOVA_03417) [32] and a GH10 endo-acting xylanase (BACOVA_04387) to generate shorter AXOS. These shorter oligosaccharides enter the cytoplasm through a major facility superfamily (MFS) transporter (BACOVA_04388) to their final degradation by a GH3 β-glucosidase (BACOVA_04319). AXOS is recognized by the hybrid two-component system (HTCS) sensor-regulator (BACOVA_03437), upon which they stimulate the transcription of the gene cluster.

The more complex corn GAX induces significant activation of PUL-XylL but less extent of PUL-XylS [68]. The extracellular binding and degradation of GAX requires SGBP, SusD-like and GH3 β-xylosidase [32], GH30 glucuronoxylanase, GH43 arabinofuranosidase, and GH98 endo-xylanase enzymes. XOS is degraded by the cytoplasmic GH31 α-xylosidase, GH43 arabinofuranosidase [32], GH95 α-L-galactosidase, and GH115 α-glucuronidase [72].

For *P. bryantii*, the gene products of two gene clusters *xusABCD* and *xynABCDER* are involved in xylan degradation. The former gene cluster plays an important role in xylan binding, degrading, and transporting XOS into the periplasmic space [73,74]. Being homologous to *B. thetaiotaomicron* SusD, both XusB and XusD can bind to extracellular xylan. XynC (a GH10 xylanase) has a type II N-terminal signal peptide and is, therefore, likely located on the outer membrane [75]. XynC and XynE (a putative esterase) cooperate in degrading xylan into XOS, which are transported across the outer membrane and into the periplasmic space via the TonB transporter XusA/XusC [34]. The latter gene cluster encodes the endo-xylanase XynA and β-xylosidase XynB, which are predicted to further degrade the XOS in the periplasm [74]. The products are then transported into the cytoplasm by the inner membrane transporter XynD [73]. The smaller XOS is further broken down into simple sugars by enzymes resident in the cytoplasm. XynR is an HTCS required to induce expression of the xylan utilization genes through a series of signal cascade reactions [76].

### 3.3. Mannan Utilization System (Mannan-PUL)

Mannan is another kind of hemi-cellulosic plant cell wall polysaccharide rich in feedstuffs such as palm kernel cake, soybean meal, and rapeseed meal. Galactomannan such as locust bean gum (LBG), abundant in the seeds of leguminous plants [77,78], is often used as a model mannan substrate in studying mannan degradation and utilization. LBG has a β(1,4)-d-mannan backbone with d-galactose sugar side chains [79]. The *B. ovatus* strain ATCC 8483 contains a PUL (*Bo*ManPUL) specific for mannan utilization [33,80]. In the *Bo*ManPUL gene cluster, two GH26 endo-mannanases (*Bo*Man26A and *Bo*Man26B), a GH36 α-galactosidase *Bo*Gal36A, and a SusD-like surface glycan-binding protein are essential for utilization of galactomannan [33,81]. The intact, highly polymerized galactomannan molecule is held and positioned on the *B. ovatus* cell membrane by the surface protein SusE and degraded into large fragments by the surface β-1,4-mannanase *Bo*Man26B, which are imported by the SusC/SusD-like complex [33] into the periplasmic space for further processing. The galactomanno-oligosaccharides are degalactosylated by the GH36 family α-galactosidase *Bo*Gal36A [80] and subsequently hydrolyzed into mannobiose by the β-mannanase *Bo*Man26A [33]. The inner-membrane associated symporter protein transports mannobiose into the cytosol for metabolism. The HTCS-like regulator is responsible for stimulating the transcription of the gene cluster upon growth of the bacterium in a mannan medium [28].

### 3.4. Pectin Utilization System (Pectin-PUL)

Pectin is most abundant in the middle lamellae of plant cell wall. Structurally different types of pectin are classified as homogalacturonan (HG), rhamnogalacturonan-I (RGI), and rhamnogalacturonan II (RG-II) [75]. HG has a rather simple structure as α-1,4-linked-d-galacturonic acid polymer, but the RGI backbone is instead composed of repeating [→α-d-galacturonic acid-1,2-α-l-rhamnose-1,4←]_n_ units. In addition, the RGI backbone is substituted by varying forms of galactan or arabinan in different plants [82,83,84]. RGII represents the most complex pectin by comprising 13 different types of sugars and 21 different glycosidic linkages. The gut bacterium *B. thetaiotaomicron* can utilize all types of pectin mediated by the PUL systems [28] which involve a total of 30 GHs and PLs [85]. The PULs associated with pectin degradation are specific for arabinan (BT0348-BT0369, Ara-PUL), galactan (BT4667-BT4673, Gal-PUL), RGI backbone (BT4145-BT4183, RGI-PUL), and HG (BT4108-BT4124, HG-PUL) [28].

For arabinan, the PUL encodes two pairs of SusC and SusD homologues (SusC_H_/SusD_H_, BT0362/BT0361 and BT0364/BT0363) which may increase the chance of the bacterium to capture pectin [28]. Arabinan binds to the cell surface through action of the SGBP BT0365. The surface GH43 endo-acting α-1,5-arabinanases (BT0360 and BT0367) cleave both linear and decorated arabinans into arabino-oligosaccharides [85,86], which are transported into the periplasm through the SusC_H_/SusD_H_ complex. Periplasmic degradation of arabino-oligosaccharides is carried out by three exo-α-l-arabinofuranosidases: BT0368, BT0369, and BT0348. Specifically, BT0369 removes the α-1,2-l-arabinofuranosic side chain, while the GH51 enzymes BT0348 and BT0368 likely cleave α-1,3-arabinofuranosyl linkage and the α-1,5-arabinofuranosyl linkage, respectively. As a result, arabinose can be readily released by the β-1,2-L-arabinofuranosidase BT0349 and enter the cell for further metabolic utilization [86]. The HTCS (BT0366) is responsible for transcriptional activation of the whole gene cluster upon recognizing the linear arabinan-derived oligosaccharides [85].

For galactan degradation, the SGBP BT4669 binds the polysaccharide and the GH53 endo-β-1,4-galactanase (BT4668) depolymerizes it to galacto-oligosaccharides [37]. Transported by SusC and SusD homologues (SusC_H_/SusD_H_, BT4670/BT4671) into the periplasm, the oligosaccharides are further degraded by the GH2 β-1,4-galactosidase (BT4667) into galactose, which enters the cell for metabolism. Transcription of the genes in the cluster is regulated by the HTCS sensor (BT4673), which recognizes small galacto-oligosaccharides [85].

The HG molecule is bound to the bacterial cell surface by the SGBP BT4112 and degraded by the pectate lyases BT4116 and BT4119. The oligosaccharide products are transported into the periplasm through the SusC_H_/SusD_H_ transporter complexes (BT4114/BT4113 and BT4121/BT4122) and degraded by the third lyase BT4115 [85]. A GH105 unsaturated α-galacturonidase BT4108 removes 4,5-linked unsaturated galacturonic acid from HG-derived oligosaccharides, generating products which can activate the HTCS system (BT4124). The products are finally depolymerized to galacturonic acid by the GH28 exo-α-D-galacturonidase BT4123 [85].

RGI is bound by the cell surface protein SGBP (BT4167) for degradation by the outer membrane lyase BT4170. The RGI-derived oligosaccharides are transported into the periplasm via the two SusC_H_/SusD_H_ transporter complexes (BT4164/BT4165 and BT4168/BT4169). These oligosaccharides released from pectin contain remnants from arabinan, galactan, and HG which must be removed for subsequent degradation of the backbone. In the periplasm, the galactan substituent is cleaved by synergistic action of three exo-β-1,4-galactosidases: BT4151, BT4156, and BT4160 [87]. The esterase BT4158 releases the acetyl group from D-galacturonic acid of the oligosaccharides, the GH2 β-glucuronidase BT4181 cleaves the single d-glucuronic acid, the GH28 α-d-galacturonidase BT4155 cleaves the rhamnosidic linkage, and the GH27 α-d-galactosidase BT4157 likely targets single α-galactose [85,88]. The synergistic action of these enzymes makes RGI-oligosaccharides simpler and thus the linear oligosaccharides chain is further cleaved by the lyases BT4175 and BT4183, generating 4,5-unsaturated galacturonic acid. The unsaturated residues are removed from RGI-oligosaccharides by two galacturonidases (BT4176 and BT4174) specific to unsaturated groups. Finally, the products are completely degraded through the successive action of an RGI-specific GH106 α-l-rhamnosidase (BT4145) and three GH28 rhamnogalacturonidases (BT4146, BT4153, and BT4149). The end products l-rhamnose, d-galacturonic acid, and d-glucuronic acid are transported into the cell through the transporters on the inner membrane [85].

## 4. PULs Are Associated with Two Common Strategies for Bacteroidetes to Utilize PCWP

With co-regulated enzymes cooperating to capture and degrade PCWP and import the oligosaccharides, PULs enable Bacteroidetes to utilize PCWP highly efficiently and thrive in the very competitive intestinal environment. Being highly abundant, Bacteroidetes is physiologically connected with other gut microbes [89]. Although PULs are functionally much diversified, the Bacteroidetes members basically assimilate PCWPs through two common strategies: one is selfish while the other is intraspecific cooperative sharing. Both two strategies contribute to sustain the balance of the entire intestinal micro ecosystem, enabling not only Bacteroides to gain growth competence, but also other microbes in the gut community to acquire essential carbon sources and energy for survival.

Mannan utilization by *B. thetaiotaomicron* stands as a good example of the first strategy: most large PUL-produced oligosaccharides are transported into the periplasm and further degraded therein but not on the cell surface. This prevents other bacterial species from using partially degraded products [90]. In contrast, for the latter strategy, part of the degradation products are released and can be used by other bacterial species in the gut community [91,92]. The recipient cells release metabolites, some of which are able to in turn increase the viability of the primary degrader. This cooperative evolution among different species forms a network of PCWP utilization, helping to establish organized ecological cooperative units in the intestinal tract and promote the adaptability of the entire gut microbial community. For example, the *B. ovatus* outer membrane enzymes (Inulin-PUL, BACOVA_04502, and BACOVA_04503) depolymerize inulin extracellularly which can be used by *B. vulgatus* by cross-feeding. In return, *B. vulgatus* increases the adaptability of *B. ovatus*, possibly by detoxifying inhibitory substances or providing growth-promoting factors [91,93].

In cross-feeding, the oligosaccharides released by the primary degrader do not necessarily promote the growth of every microorganism in the gut community, but instead selectively affect specific species, depending on the types of oligosaccharide products. *B. thetaiotaomicron* releases a range of complex oligosaccharides (galacto-oligosaccharides, arabinao-oligosaccharides, HG-oligosaccharides, RGI-oligosaccharides, and so on) from pectin into the culture medium, which are able to support the growth of *B. uniformis* without the ability to degrade galactan. However, co-culturing of *B. thetaiotaomicron* cannot promote growth of *Bacteroides massiliensis*, which can utilize HG- or RGI-oligosaccharides. It was hypothesized that *B. thetaiotaomicron* releases pectin oligosaccharides with degrees of polymerization larger than those suitable for *B. massiliensis* utilization [85]. Preferential degradation of some glycans over the others is likely to play a central role in shaping the complex relationships of the gut microbiota [94,95,96].

## 5. Conclusions

The ever-changing gut microbiota has been repeatedly proven to be associated with the health status of animals. Intended modulation of the microbial composition in the gut may, therefore, benefit the hosts. Some *Bacteroides* spp. have been proven to be beneficial, rendering them potential targets of intended modulation. The modulator molecules are often small chemicals (for example, Ganoderma meroterpene derivative for *Bacteroides xylansolvens* [97]); however, with the knowledge in PUL-mediated plant cell wall polysaccharides utilization, it is reasonable that molecules associated with the PUL system can also be used to modulate the gut microbiota. This is based on the facts that in Bacteroidetes, selective PUL-mediated PCWP degradation produces oligosaccharides, serving as the carbon source for and supporting growth of specific bacteria; that Bacteroidetes are physiologically networking with other species in the gut microbial community; and that close coordination among community members are present. For example, *B. thetaiotaomicron* has very limited ability to utilize resistant starch, and must rely on *Ruminococcus bromii* and other related bacteria to perform the initial degradation of particulate starch [98,99].

PCWPs, known as a form of dietary fiber, are the first type of molecules associated with the PUL system and useful in modulating the gut microbiota. The Bacteroidetes PUL system components (glycoside hydrolase and polysaccharide lyase, transporters, SGBPs, and HTCSs) all have preference for certain kinds of polysaccharides or their degradation derivatives (oligosaccharides), suggesting that foods/feeds rich in these PCWPs will stimulate growth of specific bacteria in the gut.

Enzymes (GHs or PLs), on the other hand, are also candidate molecules targeting the PUL systems towards gut microbiota modulation. In animals, the exogenously added enzymes, such as cellulase and xylanase, degrade PCWPs into oligosaccharides stimulating growth of specific Bacteroiedes species. The acting modes of enzymes vary according to their CAZy families and catalysis mechanisms, leading to different patterns of the released oligosaccharides, which preferentially support the growth of certain microbes.

Furthermore, the strategy used by the PUL systems can also be exploited for gut microbiota manipulation. Note that one important feature of the PUL system is the proximity of the enzyme to the bacterial cell surface, which ensures that the oligosaccharides are rapidly assimilated by the bacteria. Although enzymes are routinely added in animal feeds, both the enzymes and the released oligosaccharides are evenly distributed, limiting the selectivity in stimulating growth of specific bacteria. Therefore, it is expected that, if the exogenously added enzymes can be engineered to mimic the PUL counterparts and bind to specific bacterial cells, the local concentration of released sugars should be higher and the bacteria may have higher competence in the gut microbial community.

## Figures and Tables

**Figure 1 ijms-22-03077-f001:**
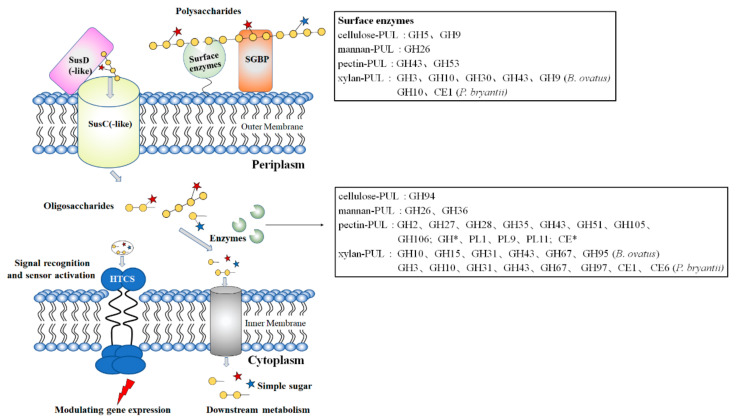
Schematic diagram of the Bacteroides polysaccharide utilization systems (PULs). Polysaccharide molecule is bound to a SGBP on the Bacteroides cell membrane surface and degraded into large fragments by the action of surface enzymes. Then the oligosaccharides, which are transported across the outer membrane and into the periplasmic space via the TonB-dependent transporter SusC/SusD-like for further processing. The oligosaccharides are subsequently hydrolyzed into simple sugar by the enzymes located in the cytoplasm space. The inner-membrane associated symporter protein transports the products into the cytosol for metabolism. The HTCS-like regulator is required to induce expression of the polysaccharide utilization genes through a series of signal cascade reactions. GH, glycoside hydrolase; PL, polysaccharide lyase; CE, carbohydrate esterase; *, new family. Yellow circles stand for the monosaccharides that form the backbone of a polysaccharide. The red and blue stars represent different groups on the side chain of the polysaccharides.

**Table 1 ijms-22-03077-t001:** List of selected Bacteroidetes PUL systems and their encoded GH enzymes.

Species	Polysaccharide Utilization Systems (PULs)	GH Family Number	Activities in Family
Bacteroidales phylotype AC2a	cellulose-PUL	GH94	cellobiose phosphorylase (EC 2.4.1.20)
		GH5	cellulase (EC 3.2.1.4)
		GH9	cellobiohydrolase (EC 3.2.1.91)
*B. ovatus* ATCC 8483	xylan-PUL	GH10	endo-1,4-β-xylanase (EC 3.2.1.8)
		GH3	1,4-β-xylosidase (EC 3.2.1.37)/β-glucosidase (EC 3.2.1.21)
		GH43	α-L-arabinofuranosidase (EC 3.2.1.55)
		GH30	glucuronoarabinoxylan endo-β-1,4-xylanase (EC 3.2.1.136)
		GH98	endo-β-1,4-xylanase (EC 3.2.1.8)
		GH31	α-xylosidase (EC 3.2.1.177)
		GH95	α-l-galactosidase (EC 3.2.1.-)
		GH115	xylan α-1,2-glucuronidase (3.2.1.131)
*P. bryantii*	xylan-PUL	GH10	endo-1,4-β-xylanase (EC 3.2.1.8)
*B. ovatus* ATCC 8483	mannan-PUL	GH36	α-galactosidase (EC 3.2.1.22)
		GH26	β-mannanase (EC 3.2.1.78)
*B. thetaiotaomicron*	pectin-PUL	GH43	endo-α-1,5-L-arabinanase (EC 3.2.1.99)
		GH51	α-L-arabinofuranosidase (EC 3.2.1.55)
		GH2	β-galactosidase (EC 3.2.1.23)
		GH53	endo-β-1,4-galactanase (EC 3.2.1.89)
		GH2	β-glucuronidase (EC 3.2.1.31)
		GH28	rhamnogalacturonan α-1,2-galacturonohydrolase (EC 3.2.1.173)
		GH27	α-galactosidase (EC 3.2.1.22)
		GH106	α-L-rhamnosidase (EC 3.2.1.40)
		GH28	rhamnogalacturonan α-1,2-galacturonohydrolase (EC 3.2.1.173)

## Data Availability

All data supporting the conclusions of this article are included within the manuscript.

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
