# Peer review of "PUL-Mediated Plant Cell Wall Polysaccharide Utilization in the Gut Bacteroidetes"

_ijms, 2021, doi:10.3390/ijms22063077_

Round 1

Reviewer 1 Report

This is a review of a manuscript entitled “PUL-mediated plant cell wall polysaccharides utilization in the gut Bacteroidetes” authored by Hao et al. In this review, the authors summarize recent advances in our knowledge of how gut bacteria from the phylum, Bacteroidetes degrade and utilize complex polysaccharides. This timely review will serve as an excellent resource for those investigators interested in understanding the mechanisms for polysaccharide utilization in the gut. The review is well-written and concise. I am in favor of publication following attention to the following minor comments:

Minor comments:

1) In the title “polysaccharides” should be “polysaccharide”

2) Abstract, line 11 and elsewhere in the manuscript: Change “destruction” to “deconstruction”

3) Page 1, line 34: Consider adding a sentence to mention the impact of microbial polysaccharide metabolism on the host before “PCWP degradation requires…” One option to consider is: “These microbes in turn convert PCWPs to short chain fatty acids that influence host physiology and health.”

4) Page 2, line 50: Insert “outside of the” before “cell membrane”

5) Page 2, line 61: Replace “mainly co-localized” with “are typically co-localized within the genome”

6) Page 2, line 61: Remove the “.” after [36]

7) Page 2, line 63: Replace “strictly” with “likely”

8) Figure 1: Include a legend defining the circles and stars

9) Figure 1: “Activating gene expression” should be “Modulating gene expression”

10) Page 2, line 72: replace “does not draw” with “has not drawn”

11) Page 3, line 99: replace “needs” with “requires the”

12) Page 3, line 103: All species names should be italicized. Also, after spelling out the first genus name, subsequent names can use the first letter followed by a period. For example the next organism should be B. fragilis.

13) Page 4, line 136: Insert “likely” before “located” because this localization has not been demonstrated experimentally.

Reviewer 2 Report

This manuscript is a review summarizing the progress made in polysaccharide utilization systems (PUL) that specifically degrade plant cell wall polysaccharides to be utilized by bacteria and their hosts. The manuscript represents a nice summary of the history in the field. However, the manuscript would be strengthened with the inclusion of a section about the Bacteroidetes. For example, how they were discovered, how are they studied/analyzed (the approaches), what are the bottleneck in the field. Another section about the potential applications would be a plus. For example, how to manipulate diet to improve health or eliminate some diseases…etc.

Also, including a table listing some important Bacteroidetes and PUL (linked to GHs in CAZY database) would make the manuscript more informative.

Minor comments:

1-The font in Figure 1 is too small. Legend could include more details to make it more self-explanatory.

2-Some terms need to be defined for people not working in the field. For example, what is the difference between Bacteroidetes and bacteria…..
